# In Situ Synthesis of CsPbX_3_/Polyacrylonitrile Nanofibers with Water-Stability and Color-Tunability for Anti-Counterfeiting and LEDs

**DOI:** 10.3390/polym16111568

**Published:** 2024-06-01

**Authors:** Yinbiao Shi, Xiaojia Su, Xiaoyan Wang, Mingye Ding

**Affiliations:** 1College of Science, Nanjing Forestry University, Nanjing 210037, China; bdd@njfu.edu.cn (Y.S.); 2211110417@njfu.edu.cn (X.S.); wangxiaoyan@njfu.edu.cn (X.W.); 2Key Laboratory of Advanced Energy Materials Chemistry (Ministry of Education), Nankai University, Tianjin 300071, China; 3College of Engineering and Applied Sciences, State Key Laboratory of Analytical Chemistry for Life Science, National Laboratory of Micro-Structures, Nanjing University, Nanjing 210023, China

**Keywords:** perovskite quantum dots, electrospinning, nanofibers, multicolor tunable emissions, water stability

## Abstract

Inorganic CsPbX_3_ (X = Cl, Br, I) perovskite quantum dots (PQDs) have attracted widespread attention due to their excellent optical properties and extensive application prospects. However, their inherent structural instability significantly hinders their practical application despite their outstanding optical performance. To enhance stability, an in situ electrospinning strategy was used to synthesize CsPbX_3_/polyacrylonitrile composite nanofibers. By optimizing process parameters (e.g., halide ratio, electrospinning voltage, and heat treatment temperature), all-inorganic CsPbX_3_ PQDs have been successfully grown in a polyacrylonitrile (PAN) matrix. During the electrospinning process, the rapid solidification of electrospun fibers not only effectively constrained the formation of large-sized PQDs but also provided effective physical protection for PQDs, resulting in the improvement in the water stability of PQDs by minimizing external environmental interference. Even after storage in water for over 100 days, the PQDs maintained approximately 93.5% of their photoluminescence intensity. Through the adjustment of halogen elements, the as-obtained composite nanofibers exhibited color-tunable luminescence in the visible light region, and based on this, a series of multicolor anti-counterfeiting patterns were fabricated. Additionally, benefiting from the excellent water stability and optical performance, the CsPbBr_3_/PAN composite film was combined with red-emitting K_2_SiF_6_:Mn^4+^ (KSF) on a blue LED (460 nm), producing a stable and efficient WLED device with a color temperature of around 6000 K and CIE coordinates of (0.318, 0.322). These results provide a general approach to synthesizing PQDs/polymer nanocomposites with excellent water stability and multicolor emission, thereby promoting their practical applications in multifunctional optoelectronic devices and advanced anti-counterfeiting.

## 1. Introduction

Recently, inorganic cesium lead halide perovskite quantum dots (PQDs) with the molecular formula CsPbX_3_ (X = Cl, Br, I) have undergone extensive research as promising next-generation optoelectronic materials due to their high photoluminescent quantum yield (PLQY), color-tunable luminescence, narrow emission bands and so on [1,2,3,4]. They have gained significant attention in applications such as anti-counterfeiting, photocatalysis, bioimaging, displays and lighting [5,6]. However, the instability of PQDs can be attributed to the lower formation energy and ionic character of chemical bonds, which render them highly sensitive to environmental factors (e.g., water, temperature or radiation), greatly limiting their practical applications in optoelectronics [7,8,9]. To enhance the stability of PQDs, a series of encapsulation strategies, including PQDs/mesoporous silica [10], PQDs/metal–organic frameworks [11] and PQDs/glass composites [12], have been proposed. However, the synthesis and processing processes of these methods are complex and cumbersome, including mesoporous template synthesis, sol–gel preparation, sintering, etc., and are not suitable for large-scale fabrication. Electrospinning technology has garnered significant attention in recent years. In addition to its application in preparing basic films, it has been demonstrated as a simple and high-throughput process for producing polymer nanofibers with enhanced crystallinity [13,14]. Additionally, electrospun nanofibers can serve as effective substrates for functional materials such as metal nanoparticles and quantum dots. The spatially confined effect of electrospun nanofibers can effectively enhance the stability of quantum dots and suppress the occurrence of aggregation phenomena [15], making it a feasible technique for manufacturing perovskite nanofiber films. For example, Fan’s group utilized a one-step emulsion electrospinning method to produce organic–inorganic hybrid perovskite nanocrystals/polymer nanofibers, exhibiting excellent stability even in humid conditions (>60%) and underwater [16]. Similarly, Chen’s group employed electrospinning in the one-step fabrication of uniform luminescent nanofibers based on a perovskite core–shell structure, with the perovskite and polymer serving as the core and shell, respectively. These nanofibers retained 50% of their initial luminescent intensity after immersion in water for 48 h [17]. The aforementioned similar studies have indeed enhanced the stability of PQDs to some extent. However, their long-term stability in water fails to meet the requirements for underwater background applications.

In this study, the selection of polyacrylonitrile (PAN) as the polymer matrix is due to its excellent weather resistance, UV resistance and good water/thermal stability [18,19,20]. Furthermore, polymer fibers also have the advantages of having a high surface specific surface area, adjustable diameter and uniform topography [15,21]. Hence, CsPbX_3_/PAN composite nanofibers have been successfully synthesized via an in situ electrospinning method, aiming to achieve the protection and preservation of CsPbX_3_ PQDs. Furthermore, a systematic investigation was conducted on the influence of electrospinning parameters (e.g., applied voltage, halide ratio and heat treatment temperature) on the morphology, optical properties and chemical stability of CsPbX_3_/PAN composite nanofibers. The advantage of this process lies in the rapid stretching and solidification of PAN nanofibers during the electrospinning process, which effectively restricts the generation of large-sized PQDs and provides strong protection for CsPbX_3_ PQDs inside PAN polymers. Thus, the electrospun CsPbBr_3_/PAN composite film maintained long-term high stability in water, retaining over 93.5% of the photoluminescence intensity for more than 100 days. In addition, the composite fiber film exhibited excellent multicolor luminescence (a tunable emission spectrum from blue to red and high color purity) by adjusting the proportion of the halogen element. Considering the outstanding stability and luminescence performance, combined with the simple and efficient electrospinning process that integrated CsPbX_3_ PQDs with PAN nanofibers, a novel composite film material was formed. This significantly enhances the water and thermal stability of PQDs and provides new approaches and methods for their practical applications in advanced anti-counterfeiting and white LEDs.

## 2. Experimental Section

### 2.1. Materials and Chemicals

Lead bromide (PbBr_2_, ≥99.0%), lead chloride (PbCl_2_, ≥99.99%), lead iodide (PbI_2_, ≥98.0%,), cesium bromide (CsBr, ≥99.5%), cesium chloride (CsCl, ≥99.5%), cesium iodide (CsI, ≥99.5%), N,N-Dimethylformamide (DMF, ≥99.9%), dimethyl sulfoxide (DMSO, ≥99.8%) and polyacrylonitrile (PAN, Mw ≈ 150,000) were purchased from Aladdin (Shanghai, China). K_2_SiF_6_:Mn^4+^ (KSF) phosphor was obtained from ZK HaoYe DongGuan Material Technology Co., Ltd. (Dongguan, China). All the reagents were used without further purification.

### 2.2. Preparation of PQDs/PAN Nanofibers

Typically, 0.5 mmol of PbX_2_ and 0.5 mmol of CsX (X = Br, I) were dissolved in 10 mL of DMF. Then, 1.0 g of PAN was added into the above mixture under stirring. To dissolve PbCl_2_ and CsCl, 5 mL of DMSO and 5 mL of DMF were used to prepare a precursor solution. The synthesis of various PQDs/PAN composite nanofibers was accomplished via a uniaxial electrospinning technique. A stainless-steel needle with a size of 20 G was employed during the electrospinning process. A total volume of 10 mL of precursor spinning solution was delivered using an electric pump at a rate of 2 mL/h. A fixed voltage of 15 kV was applied, with the needle positioned 15 cm away from the receiving substrate. The electrospinning process lasted for 2.5 h. Finally, the CsPbX_3_/PAN composite nanofibers were dried in an oven at 60 °C for 1 h to eliminate any residual organic solvents.

### 2.3. Fabrication of WLED

Then, 0.1 g of K_2_SiF_6_:Mn^4+^ (KSF) phosphor powder was thoroughly blended with 5 g of organosilica gel and continuously stirred for 3 h. The resulting solution mixture was carefully applied onto a blue LED chip (λ = 460 nm), which was covered with a dried CsPbBr_3_/PAN composite film. The assembly underwent curing at 80 °C for 5 h.

### 2.4. Characterizations

X-ray diffraction patterns of samples were collected by using an X-ray diffractometer (XRD, Ultima IV, Rigaku, Tokyo, Japan). Scanning electron microscopy (SEM) was performed on a JSM-7600F (JEOL, Tokyo, Japan) electron microscope. Transmission electron microscopy (TEM) and high-resolution transmission electron microscopy (HRTEM) images were taken on a JEM-2100F (JEOL, Tokyo, Japan) transmission electron microscope equipped with an X-ray spectrometer detector. The fluorescence distribution was confirmed by a Leica LSM-710 (Zeiss, Oberkochen, Germany) laser scanning confocal microscope (LSCM). UV-vis absorption spectra were recorded by a PE Lambda-950 UV-vis-NIR spectrophotometer (PerkinElmer, Waltham, MA, USA). The photoluminescence (PL) spectra and decay lifetimes were measured on an Edinburgh FLS980 fluorescence spectrometer (Edinburgh, Edinburgh, UK). The luminescent photographs were taken by a Canon EDS 70D digital camera (Canon, Tokyo, Japan).

## 3. Results and Discussions

Figure 1a depicts a schematic illustration of the one-step uniaxial electrospinning process. Inside the syringe, a PAN mixed solution containing CsX and PbX_2_ (X = Cl, Br, I) in varying proportions is injected. Subsequently, the mixed precursor solution is continuously ejected through a spinning needle (be applied with a fixed voltage) with a diameter of 0.51 mm (20 G specification). Simultaneously, a fixed negative voltage is applied to the receiving substrate to provide traction to the spinning solution, stretching it into nanofibers during solidification. Within this process (Figure 1b), the nanofibers act as miniature nanoreactors, where the crystallization of perovskite quantum dots occurs along with polymer solidification [21], and the rapid solidification of PAN during electrospinning imposes a constraint effect on quantum dots, suppressing their rapid growth in the absence of other ligands [22]. At the same time, the high voltage and large specific surface area of the nanofibers promote the rapid evaporation of DMF in a very short time [23]. In this process, the needle makes a reciprocating motion from left to right, and by adjusting the ratio between different halogens (Cl, Br, I), large-area, regular, multicomponent PQD nanofiber films can be prepared (Figure 1c–g). These CsPbX_3_/PAN composite nanofibers exhibit excellent foldability, allowing arbitrary folding and cutting, demonstrating significant potential in wearable displays and other flexible luminescent devices.

Firstly, the comprehensive morphological characterization of pure PAN and CsPbBr_3_/PAN composite nanofibers has been investigated, as shown in Figure 2. The pure PAN nanofibers exhibit uniform thickness and a smooth surface, with an average diameter of approximately 480 nm (Figure 2a,b). As illustrated in Figure 2c,d, the average diameters of CsPbBr_3_/PAN (CsBr:PbBr_2_ is 1:1) composite nanofibers are around 350 ± 50 nm, indicating that these are finer fibers compared to pure PAN nanofibers. This is due to the addition of PQDS, which affects the arrangement and accumulation of pan molecules during fiber formation. And this results in a reduction in fiber diameter. Additionally, it can be observed that the surface of the nanofibers becomes slightly rougher, which may be attributed to residues of CsBr and PbBr_2_. Furthermore, the elemental mapping of the as-obtained composite nanofibers is provided in Figure 2e. From the figure, it is evident that the distribution of C, Pb and Br elements can be clearly observed along the as-obtained nanofibers. This confirms that PQDs are able to successfully grow inside nanofibers. Laser confocal fluorescence microscopy was then utilized for the random testing of CsPbBr_3_/PAN composite nanofibers (Figure 2f,g). The images show the distribution and physical dimensions of electrospun fibers under bright-field conditions and actual fluorescence images under 385 nm excitation. Merging the images under bright-field and fluorescence conditions reveals that all fibers in CsPbBr_3_/PAN electrospun fibers exhibit clear and uniform fluorescence. Finally, an XRD characterization test was conducted on CsPbBr_3_/PAN composite nanofibers to further investigate their specific structures and compositions. As seen in Figure 2h, the characteristic diffraction peaks of CsPbBr_3_/PAN composite nanofibers align with the standard X-ray diffraction pattern of CsPbBr_3_ (PDF#18-0364) and PAN (PDF#48-2119), where the diffraction angles 2θ = 15.2°, 21.54° and 30.72° correspond to the (100), (110) and (200) crystal planes of the CsPbBr_3_ perovskite cubic phase. These angles match well with the standard CsPbBr_3_ cubic phase, and no other impurity phases are observed. The above comprehensive characterization analysis validates the feasibility of the in situ synthesis of PQDs nanofibers via electrospinning.

Next, PL properties and chemical stability are displayed in Figure 3. Figure 3a illustrates the UV-vis absorption spectra and PL spectra of the CsPbBr_3_/PAN composite nanofibers. The UV–visible absorption peak of this green film extends to around 525 nm, and it exhibits a PL peak at 512 nm with Full Width at Half Maximum (FWHM) of 24 nm. The inset in Figure 3a shows the actual emission photograph of the CsPbBr_3_/PAN composite film under 365 nm UV excitation, vividly showcasing its bright and pure green fluorescence. Afterwards, the influence of adjustments in the CsBr/PbBr_2_ ratio and applied voltage on the optical properties of PQDs/PAN composite nanofibers was investigated, employing CsPbBr_3_/PAN as a representative example. CsPbBr_3_/PAN composite nanofibers with different CsBr/PbBr_2_ ratios were prepared under a voltage of 14 KV. As depicted in Figure 3b, variations in the CsBr/PbBr_2_ ratio results in corresponding changes in the PL peak. As the CsBr/PbBr_2_ ratio increases, the PL peaks are located at 514 nm, 511 nm, 512 nm, 506 nm and 518 nm, with corresponding FWHM values of 22 nm, 28 nm, 24 nm, 22 nm and 43 nm, respectively. Moreover, both an excessively high and low CsBr/PbBr_2_ ratio can lead to a weakening of the PL intensity of sample. Additionally, under the condition where CsBr:PbBr_2_ = 1:1, the variation in the PL intensity of CsPbBr_3_/PAN composite nanofibers with different voltages is illustrated in Figure 3c. The PL intensity at 14 KV is twice that at 13 and 15 KV. However, with a further increase in voltage, the PL intensity begins to decrease. This is due to the intensified confinement growth effects in the nanofibers. At higher voltages, the polymer nanofibers solidify more rapidly, resulting in reduced growth time for PQDs within the nanofibers. This constrains the in situ growth of PQDs, leading to smaller particle sizes or limited crystallization. In order to achieve optimized process parameters, a systematic study was carried out on the halide ratios and applied voltages (Appendix A). From the comparison of emission spectra and PL intensity among various systems, it can be observed that when the CsBr:PbBr_2_ is 1:1 and 3:2, composite nanofibers prepared under specific voltages exhibit the generation of spurious emission peaks. When the ratio is 4:1, the PL intensity of samples prepared at different voltages is relatively weak. Comprehensively taking into account its optical performance, the optimal process parameters are determined to be voltage, 14 KV; CsBr:PbBr_2_ = 2:3.

Under optimal ratios and voltage conditions, chemical stability assessments were performed on the prepared CsPbBr_3_/PAN composite film. As observed in Figure 3d, with increasing temperature, the PL intensity of the prepared sample gradually decreases and quenches at 423 K, owing to the thermal quenching phenomenon. At elevated temperatures, the heightened exciton dissociation fosters direct interactions between electrons and phonons, thereby inducing fluorescence quenching. It is noteworthy that the PL emission peak position and FWHM remain essentially unchanged in the temperature range of 303–383 K (Appendix A), which ensures the color purity of the sample at high temperatures. This also reflects the superb optical performance of the CsPbBr_3_/PAN composite nanofibers. Finally, long-term observations and tests were conducted on the stability and PL intensity decay of the CsPbBr_3_/PAN composite film in water (completely immersed, with periodic water replenishment). Due to the encapsulation of PQDs within the polymeric matrix, the PL intensity of the CsPbBr_3_/PAN composite film does not exhibit a significant decline for over 100 days (Figure 3e). Combining Figure 3f,g, the position and FWHM of its PL peak remain essentially unchanged. Under 365 nm excitation, the film continues to exhibit bright green fluorescence in water over an extended period. Consequently, polymeric nanofibers prepared through uniaxial electrospinning demonstrate exceptional encapsulation protection for in situ synthesized PQDs within the fibers.

Given the excellent optical performance and chemical stability of CsPbBr_3_/PAN composite nanofibers, a mature electrospinning process strategy was employed to expand the preparation of PQDs/PAN composite nanofibers with different halogens, followed by a series of characterization tests. From Figure 4a–d, it can be seen that the nanofibers doped with four different halide elements exhibit uniform thickness. However, with the varying conductivity of the spinning solutions caused by different halide ions, there are significant differences in the average diameter of the nanofibers. The diameters of CsPbCl_3_/PAN, CsPbBr_1.5_Cl_1.5_/PAN, CsPbBr_1.5_I_1.5_/PAN and CsPbI_3_/PAN composite nanofibers are approximately 400~800 nm; 200~300 nm; 100~200 nm; and 600~1000 nm, respectively. The specific structures and compositions of them are illustrated in Figure 4e. As Cl^−^ and I^−^ gradually substitute for Br^−^, changes in lattice size and interplanar spacing lead to a gradual shift in the overall characteristic peaks of the samples [24]. Notably, when preparing CsPbI_3_/PAN composite nanofibers following the same procedure, the fluorescence emission spectrum appears chaotic (Appendix A). To address this issue, annealing treatment was conducted. From Appendix A, a significant enhancement in the red light emission peak is observed at an annealing temperature of 100 °C, and a single PL characteristic peak is achieved at 120 °C (Appendix A). The effectiveness of this method is further demonstrated by comparing the actual luminescence images before and after annealing in Appendix A. Building upon this, random site tests were conducted on the CsPbI_3_/PAN composite nanofibers treated at 120 °C by using LSCM (Appendix A). Nanofibers exhibit bright and clear red fluorescence, which similarly confirms the effectiveness of the annealing treatment. The above phenomenon is caused by the short reaction time, leading to the immature crystallization of CsPbI_3_ PQDs within freshly prepared nanofibers [25]. XRD comparative analysis in Figure 4e confirms that the characteristic diffraction peaks of CsPbI_3_/PAN align with the standard diffraction pattern of the orthorhombic crystal structure (PDF#74-1970). This affirms the large-scale formation of CsPbI_3_ PQDs within nanofibers.

The UV-vis absorption spectra of the four multicolor CsPbX_3_/PAN composite nanofibers are shown in Figure 5. The purple CsPbCl_3_/PAN composite nanofiber (Figure 5a) exhibits a PL peak at 418 nm and a UV–visible absorption peak at 410 nm. The blue CsPbBr_1.5_Cl_1.5_/PAN composite nanofiber (Figure 5b) shows a PL peak at 454 nm and a UV–visible absorption peak at 448 nm. The orange CsPbBr_1.5_I_1.5_/PAN composite nanofiber (Figure 5c) displays a PL peak at 534 nm and a UV–visible absorption peak at 472 nm. Lastly, the red CsPbI_3_/PAN composite nanofibers (Figure 5d) reveals a PL peak at 671 nm and a UV–visible absorption peak at 415 nm. By comparing the UV-vis absorption spectra of composite nanofibers composed of four different halogens, it can be observed that as the halide composition changes from Cl^−^ to I^−^, the CsPbX_3_/PAN composite nanofibers exhibit higher absorbance in the ultraviolet and visible spectral regions [26]. Subsequently, thermal stability tests were conducted on the four aforementioned composite nanofiber films. The results indicated that samples doped with halogen element iodine exhibit excellent thermal stability (Appendix A).

Figure 6a illustrates the photoluminescence (PL) spectra of all CsPbX_3_/PAN composite nanofibers, and their emitting wavelengths range from 418 to 671 nm. Unlike traditional cadmium-based QDs, the emission color of which is adjusted via the quantum size effect, the CsPbX_3_/PAN composite nanofibers show the flexible tunability of emission color via simply modifying the PQD compositions. Figure 6b presents corresponding actual luminescent photographs under 365 nm ultraviolet excitation, displaying vivid and vibrant colors. Later, time-resolved single-photon counting spectroscopy was employed. From Figure 6c, it can be observed that the average PL lifetime of composite nanofibers prepared with halide elements Br and Cl doping is smaller than that of PQD nanofiber films doped with Br and Cl. (Appendix A). This behavior aligns with PL decay patterns of PQDs reported in the literature [27,28,29]. As the halide atom radius increases at the X site (Cl^−^ = 1.81 Å, Br^−^ = 1.96 Å, I^−^ = 2.20 Å), it leads to the enlargement of the bandgap of PQDs, thereby increasing the carrier recombination rate. On the other hand, different halide elements can induce changes in the band structure of PQDs, which may affect the formation and recombination rate of electron–hole pairs, consequently influencing the fluorescence lifetime [27].The luminescent films of PQD nanofibers prepared by electrospinning differ significantly from traditional lanthanide-doped phosphors. CsPbX_3_/PAN composite nanofibers, with their narrow FWHM and tunable emission, can generate saturated, single-color luminescence [30,31,32]. The CIE coordinates, as shown in Figure 6d, illustrate the broad color gamut of these CsPbX_3_ PQDs (Appendix A). Considering the excellent micolor emission properties and exceptional stability of the electrospun composite films, these samples hold great promise for multicolored optoelectronic applications in aquatic environments in applications of counterfeit prevention and LEDs.

On the one hand, electrospinning offers the advantage of producing large-area, highly stable fluorescent nanofiber films; on the other hand, the flexibility (Figure 7d,e) of the nanofibers themselves allows for arbitrary cutting and assembling into any desired pattern, which renders them suitable for applications in anti-counterfeiting and WLED [33,34]. Thus, leveraging the foldability of nanofiber films attempts to cut multicolor CsPbX_3_/PAN composite films into different patterns (Figure 7). Figure 7a–c, respectively, show fluorescent patterns (a school badge and the letters NJFU) composed of individual multicolor CsPbX_3_/PAN composite films created by cutting and assembling CsPbX_3_/PAN composite films with various compositions under UV excitation (λ = 365 nm) in dark conditions. The excellent display performance implies potential applications in the field of multicolor anti-counterfeiting. This utilization of electrospun nanofiber films showcases promise for future advancements in multifunctional materials for security and display technologies.

Benefiting from the exceptional PL performance and good long-term stability of CsPbX_3_/PAN composite nanofibers, white light-emitting diodes (WLEDs) were successfully fabricated using CsPbBr_3_/PAN composite film and red-emitting KSF phosphors. Figure 8a illustrates the operational principles of the WLED, wherein a blue light-emitting diode (LED) emitting at a wavelength of 460 nm is selected as the excitation light source. The KSF and CsPbBr_3_/PAN composite film are combined and overlaid onto the LED. Following this, they are encapsulated and cured with organic adhesive, which results in a simplified WLED device [35]. Performance testing of the prepared WLED under a 20 mA drive current, as shown in Figure 8b, reveals distinct emission peaks in the blue, green and red spectral ranges. The inset presents actual emission photographs of the WLED device, providing a clear depiction of its ability to emit bright, pure white light. Finally, based on Figure 8c, the WLED device exhibits CIE coordinates of (0.318, 0.322), placing it within the white light region with a color temperature of approximately 6000 K [36]. Taking into account the chemical stability of the CsPbBr_3_/PAN composite film, it shows significant potential in light-emitting devices.

## 4. Conclusions

In summary, we employed an electrospinning in situ synthesis strategy to fabricate visible-light tunable CsPbX_3_ (X = Cl, Br, I)/PAN composite nanofibers with emitting wavelengths ranging from 418 to 671 nm. Through process optimization, specific parameters were set as follows: voltage: 14 KV; CsBr:PbBr_2_ = 2:3. The CsPbBr_3_/PAN nanofiber film exhibited good stability in water resistance tests. This remarkable stability is attributed to the effective encapsulation of PQDs by the polymeric matrix, which isolates them from the detrimental effects of oxygen and moisture. Even after immersion in water for over 100 days, the film maintained approximately 93.5% of its PL intensity. Building upon this, doping various halogen elements endowed it with multicolor luminescent properties. Combined with the flexibility of the composite films themselves, they can be fabricated into various shapes and multicolored anti-counterfeiting patterns. Furthermore, leveraging the narrow FWHM and tunable emission characteristics of CsPbX_3_/PAN (X = Cl, Br, I) composite nanofibers, we successfully prepared a bright white light-emitting diode (WLED) with CIE coordinates of (0.318, 0.322) and a color temperature of around 6000 K under blue LED excitation. These results not only provide a unique approach for the synthesis of PQDs but also advance their practical applications in optoelectronic lighting and display technologies.

## Figures and Tables

**Figure 1 polymers-16-01568-f001:**
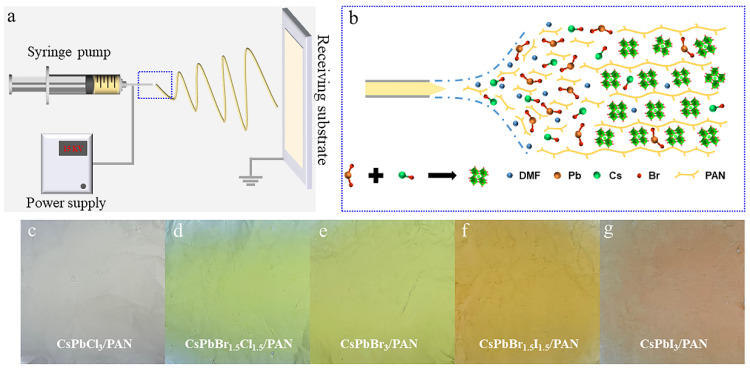
(**a**,**b**) Schematic of the one-step single-axis electrospinning setup to fabricate the perovskite light-emitting nanofibers. Photographs of as-synthesized CsPbX_3_/PAN samples: (**c**) CsPbCl_3_/PAN, (**d**) CsPbBr_1.5_Cl_1.5_/PAN, (**e**) CsPbBr_3_/PAN, (**f**) CsPbBr_1.5_I_1.5_/PAN, (**g**) CsPbI_3_/PAN.

**Figure 2 polymers-16-01568-f002:**
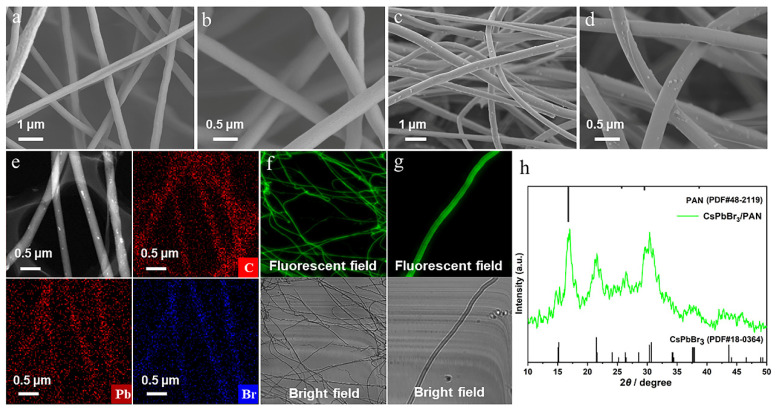
(**a**,**b**) SEM images of pure PAN nanofibers. (**c**,**d**) SEM images of CsPbBr_3_/PAN (CsBr:PbBr_2_ is 1:1) composite nanofibers. (**e**) TEM images of CsPbBr_3_/PAN composite nanofibers. EDS mapping for C, Pb and Br elements. (**f**,**g**) Fluorescent field images and bright-field images of CsPbBr_3_/PAN composite nanofibers recorded by LSCM, (**h**) XRD patterns of CsPbBr_3_/PAN composite nanofibers.

**Figure 3 polymers-16-01568-f003:**
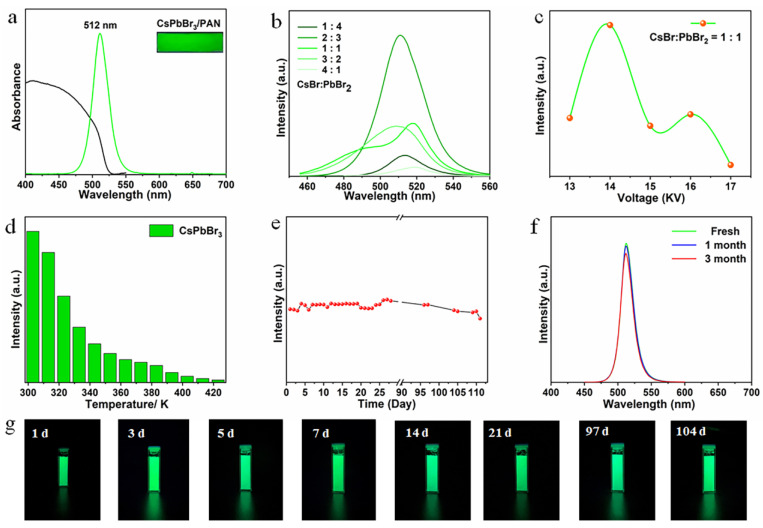
CsPbBr_3_/PAN composite nanofibers: (**a**) UV-vis absorption and PL spectra. (**b**) Different ratios of CsBr and PbBr_2_. (**c**) Different voltage. (**d**) Thermostability under 423 K. (**e**) Temporal evolution of PL intensity in room-temperature water. (**f**) Long-term storage stability. (**g**) Photos of CsPbBr_3_/PAN composite nanofibers dipped in water for 1, 3, 5, 7, 14, 21, 97 and 104 days.

**Figure 4 polymers-16-01568-f004:**
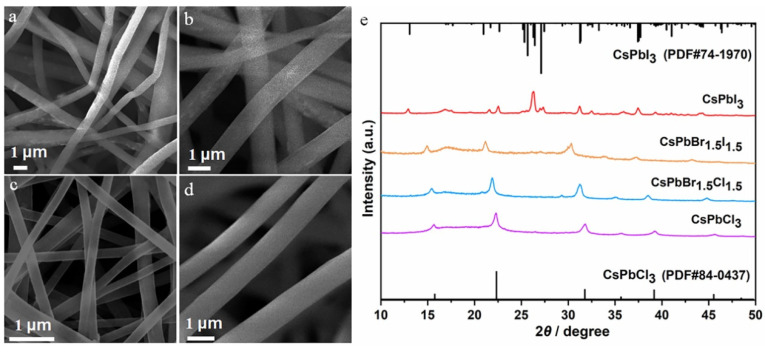
(**a**,**b**) SEM images of CsPbX_3_/PAN composite nanofibers: (**a**) CsPbCl_3_. (**b**) CsPbBr_1.5_Cl_1.5_. (**c**) CsPbBr_1.5_I_1.5_. (**d**) CsPbI_3_. (**e**) XRD patterns of CsPbX_3_/PAN composite nanofibers.

**Figure 5 polymers-16-01568-f005:**
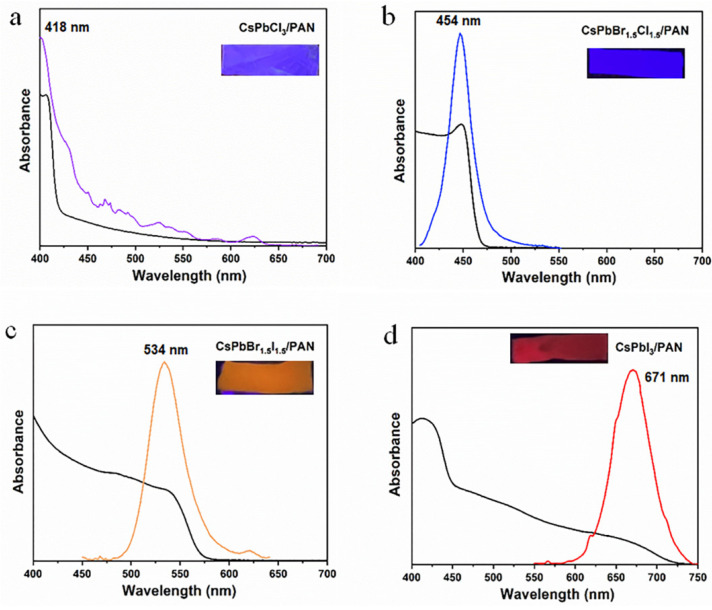
UV-vis absorption and PL spectra of CsPbX_3_/PAN composite nanofibers (**a**) CsPbCl_3_/PAN. (**b**) CsPbBr_1.5_Cl_1.5_/PAN. (**c**) CsPbBr_1.5_I_1.5_/PAN. (**d**) CsPbI_3_/PAN.

**Figure 6 polymers-16-01568-f006:**
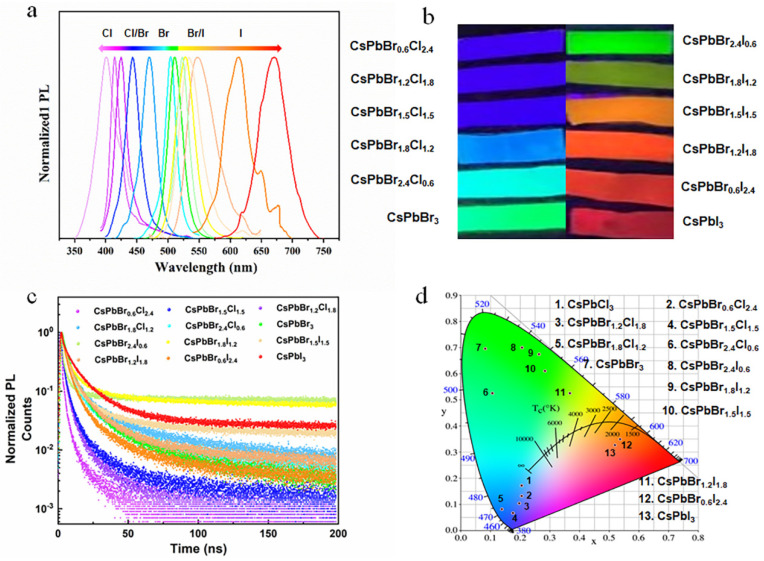
CsPbX_3_/PAN composite nanofibers with different halogen constitutions: (**a**) PL spectra. (**b**) Digital photographs under UV excitation (λ = 365 nm). (**c**) PL decay curves. (**d**) CIE coordinate and color gamut.

**Figure 7 polymers-16-01568-f007:**
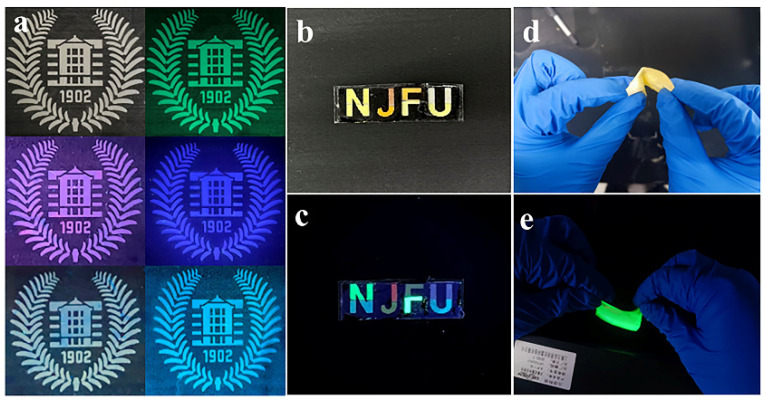
(**a**–**c**) Demonstration of the application of a multicolor fluorescent patterns for anti-counterfeiting. (**d**,**e**) Flexibility on display.

**Figure 8 polymers-16-01568-f008:**
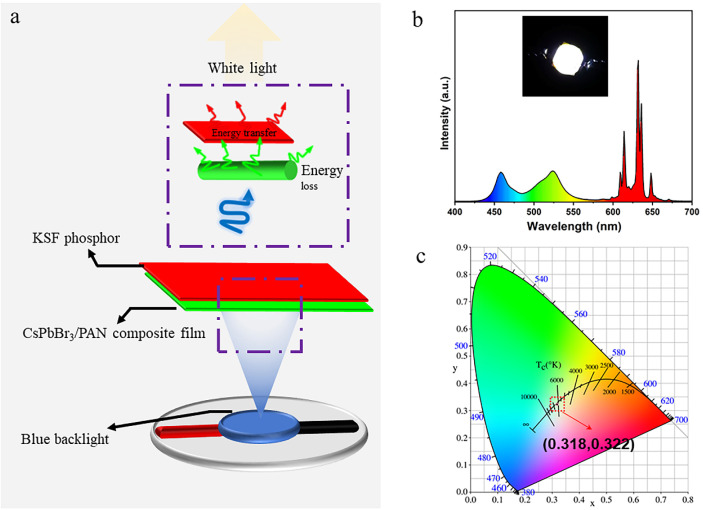
(**a**) Simplified structure of WLED. (**b**) EL spectrum (inset shows a digital camera image of the working WLED). (**c**) CIE coordinate and color gamut of WLED.

## Data Availability

Data are contained within the article and Appendix A.

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
