# Peer review of "In Situ Synthesis of CsPbX3/Polyacrylonitrile Nanofibers with Water-Stability and Color-Tunability for Anti-Counterfeiting and LEDs"

_polymers, 2024, doi:10.3390/polym16111568_

Round 1

Reviewer 1 Report

Comments and Suggestions for Authors

I find this manuscript well written and complete.

My comments and suggestions in the attached file.

Author Response

Dear editors and reviewers,

Thank you for your letter and for the reviewers’ comments concerning our manuscript entitled “In situ synthesis of CsPbX3@polyacrylonitrile nanofibers with water-stability and color-tunability for anti-counterfeiting and LEDs” (Polymers-3028831). Those comments are all valuable and very helpful for revising and improving our paper, as well as the important guiding significance to our researches. We have studied comments carefully and have made correction which we hope meet with approval. Revised portion are marked in yellow in the paper. The main corrections in the paper and the responses to the review’s comments are uploaded as an attachment.

We tried our best to improve the manuscript and made some changes in the manuscript. These changes will not influence the content and framework of the paper. And here we didn’t list the changes but marked in red in revised paper.

We appreciate for Editor/Reviewer’ warm work earnestly, and hope that the correction will meet with approval. Once again, thank you very much for your comments and suggestions.

Yours sincerely,

Mingye Ding

Reviewer 2 Report

Comments and Suggestions for Authors

Full Title: In situ synthesis of CsPbX3@polyacrylonitrile nanofibers with 2 water-stability and color-tunability for anti-counterfeiting and 3 LEDs

In the present work, Inorganic CsPbX3 (X = Cl, Br, I) perovskite quantum dots (PQDs) have attracted 13 widespread attention due to their excellent optical properties and extensive application prospects. 14 However, the inherent structural instability significantly hinders their practical application despite 15 their outstanding optical performance. . These results pro- 27 vide a general approach to synthesize PQDs/polymer nanocomposites with excellent water-stability 28 and multicolor emission, thereby promoting their practical applications in multifunctional optoe- 29 lectronic devices and advanced anti-counterfeiting. The abstract covers some results that mentioned in paper. The introduction is comprehensive and the experimental work is need discussion. Some comments could be summarized as follows:

1-     In the abstract, you should to refer on the induced changes of properties with some data.

2-     You should to refer in the introduction part, what is the reason for choosing CsPbX3/PAN composite nanofibers (it need some novelty(?

3-     The objectives are not clear – You should to mention the objective and the novelty of the work in the last paragraph of the introduction part?

4-     Why author choose fixed voltage of 15 kV was 98 applied, with the needle positioned 15 cm away from the receiving substrate? Give the practical reason for selecting these values?

5-     The rough surface is mostly important for discuss the behavior presence of hilly areas. The authors should discuss it with respect to the modifications to the surface. Please discuss the reasons for the change?

6-     Results are interesting, but should to be more discussion and comparison with other works?

7-     In Fig 1, need more discussion to refer for the these phenomena (CsPbX3/PAN composite nanofibers ex- 133 hibit excellent foldability, allowing arbitrary folding and cutting)?

8-     In result and discussion part, These angles match well 157 with the standard CsPbBr3 cubic phase, and no impurity phases are observed., need more discussion?

9-     How to measure the sample thickness?

10- Conclusions, should to be more directed toward the applications of these samples for solid-state-battery electrode Applications?

11- You must to update the refs, to be new and related of the work.

Author Response

(The authors gave the same response as above.)
